# Transcriptional signatures of somatic neoblasts and germline cells in *Macrostomum lignano*

**Magda Grudniewska[1†], Stijn Mouton[1,2,3†], Daniil Simanov[1,2,3], Frank Beltman[1], Margriet Grelling[1], Katrien de Mulder[2,3‡], Wibowo Arindrarto[2,3§], Philipp M. Weissert[1], Stefan van der Elst[2,3], Eugene Berezikov[1,2,3*]**

[1]European Research Institute for the Biology of Ageing, University of Groningen, University Medical Center Groningen, Groningen, The Netherlands; [2]Hubrecht Institute-KNAW, Utrecht, The Netherlands; [3]University Medical Centre Utrecht, Utrecht, The Netherlands

**Abstract** The regeneration-capable flatworm *Macrostomum lignano* is a powerful model organism to study the biology of stem cells in vivo. As a flatworm amenable to transgenesis, it complements the historically used planarian flatworm models, such as *Schmidtea mediterranea*. However, information on the transcriptome and markers of stem cells in *M. lignano* is limited. We generated a de novo transcriptome assembly and performed the first comprehensive characterization of gene expression in the proliferating cells of *M. lignano*, represented by somatic stem cells, called neoblasts, and germline cells. Knockdown of a selected set of neoblast genes, including *Mlig-ddx39*, *Mlig-rrm1*, *Mlig-rpa3*, *Mlig-cdk1*, and *Mlig-h2a*, confirmed their crucial role for the functionality of somatic neoblasts during homeostasis and regeneration. The generated *M. lignano* transcriptome assembly and gene expression signatures of somatic neoblasts and germline cells will be a valuable resource for future molecular studies in *M. lignano*.

**\*For correspondence:**
e.berezikov@umcg.nl

[†]These authors contributed equally to this work

**Present address:** [‡]Molecular laboratory, AZ St. Lucas Hospital, Gent, Belgium; [§]Sequencing Analysis Support Core, Leiden University Medical Center, Leiden, The Netherlands

**Competing interests:** The authors declare that no competing interests exist.

## Introduction

Flatworms are increasingly attractive models for studying biology of stem cells in vivo. These animals have an abundant population of proliferating cells, called neoblasts. Histologically, the neoblasts form a homogeneous population of small, round cells with a high nuclear/cytoplasmic ratio, which are located in the mesenchyme (*Ladurner et al., 2000*; *Baguñà, 2012*; *Rink, 2013*). The recent molecular characterization of neoblasts, however, has demonstrated that the population is heterogeneous and includes different types of progenitors and pluripotent stem cells (*Wagner et al., 2011*; *van Wolfswinkel et al., 2014*; *Tu et al., 2015*). Furthermore, it has been shown that neoblasts are the only proliferating somatic cells which are able to produce all cell types of the worm (*Morita and Best, 1974*; *Ladurner et al., 2000*; *Wagner et al., 2011*; *Baguñà, 2012*; *Rink, 2013*). Therefore, neoblasts drive a continuous cell renewal during homeostasis and produce new cells during growth and regeneration (*Ladurner et al., 2000*; *Oviedo et al., 2003*; *Takeda et al., 2009*; *González-Estévez et al., 2012*).

The most frequently used models for research on all aspects of neoblast biology are the planarians *Schmidtea mediterranea* and *Dugesia japonica* (*Reddien and Sánchez Alvarado, 2004*; *Shibata et al., 2010*; *Rink, 2013*). Phylogenetic relations within flatworms (*Laumer et al., 2015*) and with Xenacoelomorpha – the early-branching bilaterians that also have regenerative capacity (*Cannon et al., 2016*; *Hejnol and Pang, 2016*), are now well understood, paving way for studies on the neoblast origin and evolution of regeneration (*Srivastava et al., 2014*; *Gehrke and Srivastava,*

*2016*). These comparative studies will benefit from additional non-planarian flatworm models, and a basal flatworm *Macrostomum lignano* (Macrostomorpha), a marine, non-self-fertilizing hermaphrodite (*Figure 1A*) is being developed as one of such models (*Ladurner et al., 2005*). The animals are small, about 1 mm long, transparent, and easy to culture, as adults lay about one single-cell egg each day when cultured at 20°C. Worms are able to regenerate missing body parts anteriorly, posteriorly, and laterally, although the presence of the brain and pharynx is obligatory (*Egger et al., 2006*). The neoblasts are located in two lateral bands, starting from the region of the eyes and merging in the tail plate (*Figure 1A*). Besides the somatic neoblasts, proliferating cells are also present in the gonads (*Ladurner et al., 2000*). Several techniques are developed for *Macrostomum*, including antibody labeling, in situ hybridization (ISH), RNA interference (RNAi), and gene expression analysis (*Ladurner et al., 2000*, *2005*; *Pfister et al., 2007*; *De Mulder et al., 2009*; *Arbore et al., 2015*; *Plusquin et al., 2016*). Recently, the first genome and transcriptome assemblies were

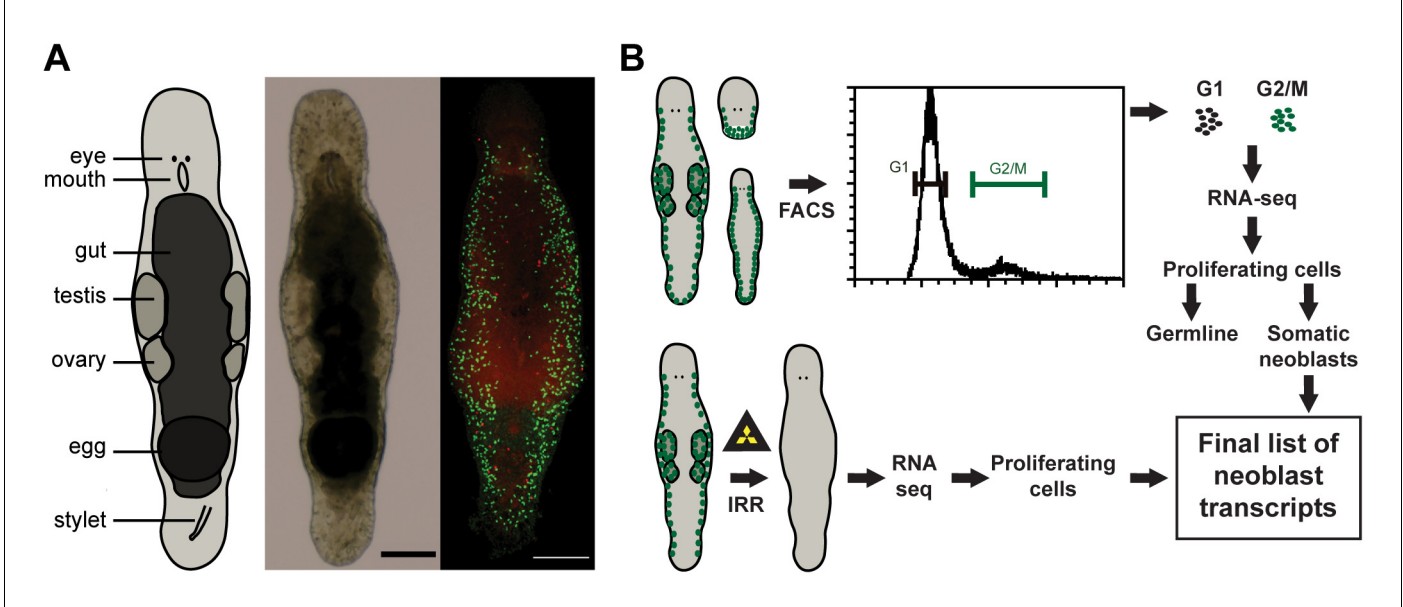

**Figure 1.** *Macrostomum lignano* as model organism and experimental set up. (**A**) Schematic representation, bright field image, and confocal projection of BrdU and phospho-histone H3 immunostaining (green: S-phase cells, red: mitotic cells) of an adult *M. lignano*. (**B**) Schematic representation of the experimental setup. Scale bar 100 μm.

The following source data and figure supplements are available for figure 1:

**Source data 1.** TransRate contigs scores for MLRNA150904 transcriptome assembly.

**Source data 2.** Gene counts, fold changes and FDR for various gene expression comparisons, and classification of genes into categories.

**Source data 3.** GO term enrichment analysis in various gene lists.

**Source data 4.** Enrichment of *S. mediterranea* and human markers in various transcript sets.

**Figure supplement 1.** Approach used to generate the *Macrostomum lignano* de novo transcriptome assembly MLRNA150904.

**Figure supplement 2.** Characteristics of MLRNA150904 transcriptome assembly.

**Figure supplement 3.** Effects of γ-irradiation on *Macrostomum lignano*.

**Figure supplement 4.** Isolation of *M. lignano* proliferating cell by fluorescence activated cell sorting (FACS).

**Figure supplement 5.** Classifications of overlaps between *M. lignano* genes and *S. mediterranea* and human homologs.

published (*Wasik et al., 2015*), and transgenesis utility was demonstrated (*Marie-Orleach et al., 2014*, *2016*). Despite this available toolbox, the described molecular markers for proliferating cells in *M. lignano* are still limited to *piwi* and *vasa*, which are expressed in both somatic neoblasts and proliferating germline cells (*Pfister et al., 2007*, *2008*; *De Mulder et al., 2009*; *Zhou et al., 2015*). Consequently, there is an urgent need to identify more useful neoblast markers to develop this animal as a model for in vivo stem cell biology.

In this paper, we present a molecular characterization of the proliferating cells of *M. lignano*. We first generated a de novo transcriptome assembly of *M. lignano*. Next, we used two approaches to identify genes specifically expressed in proliferating cells: (i) comparisons of gene expression in irradiated worms, devoid of proliferating cells, and control worms, and (ii) in FACS-isolated differentiated and proliferating cells (*Figure 1B*). Moreover, by isolating cells from adult animals, juveniles and from amputated heads, which lack germline, we could distinguish the enrichment of transcripts in the gonads and in the somatic neoblasts. As a last step, we performed an RNAi screen, revealing five conserved genes crucial for the functionality of somatic neoblasts during homeostasis and regeneration.

## Results

### De novo transcriptome assembly

To produce a comprehensive de novo transcriptome assembly of *M. lignano*, we made 22 RNA sequencing libraries using different approaches, and generated in total more than one billion sequencing reads. The data included normalized 454 library, strand-specific polyA-enriched and RiboMinus-depleted Illumina libraries, 5'-enriched RAMPAGE libraries (*Batut et al., 2013*), and 3'-specific CEL-seq libraries (*Hashimshony et al., 2012*) sequenced using the T-fill method (*Wilkening et al., 2013*), which allows exact mapping of mRNA polyadenylation sites. In order to maximize the chances of reconstructing full-length transcripts, the data were assembled using four different de novo transcriptome assemblers, results merged and re-assembled with CAP3, requiring consistent positioning of paired reads (*Figure 1—figure supplement 1*). The set of 60,180 primary transcripts, which can explain more than 90% of all sequencing reads, was designated as MLRNA150904 transcriptome assembly and used in the subsequent analyses in this study.

To assess the quality of the transcriptome assembly, we used TransRate – a recently developed reference-free approach that can detect common transcriptome assembly artefacts, such as chimeras and incomplete assembly, and provide individual transcript and overall assembly scores (*Smith-*

**Table 1.** Properties of MLRNA150904 transcriptome assembly.

| | |
|---|---|
| Number of transcripts | 60,180 |
| Total length, nt | 95,589,662 |
| Average transcript length, nt | 1588 |
| Shortest transcript, nt | 100 |
| Longest transcript, nt | 29,807 |
| CEG homologs* | 247 out of 248 |
| Transspliced transcripts | 6167 |
| TransRate score[†] | 0.4367 |
| Good TransRate transcripts | 86% |
| Human homolog genes | 8458 |
| PFAM domains | 3503 |
| *S. mediterranea* cell type-specific gene homologs[‡] | 1697 |

*Core Eukaryotic genes according **Parra et al. (2009)**.
[†]Assembly quality score according to **Smith-Unna et al. (2016)**.
[‡]Cell-type-specific genes from **Wurtzel et al. (2015)**.

*Unna et al., 2016*). The TransRate assembly score for the MLRNA150904 assembly is 0.4367 (*Table 1*, *Figure 1—figure supplement 2A*), which ranks it as the seventh highest scoring de novo transcriptome assembly out of 155 publicly available transcriptomes analyzed in *Smith-Unna et al. (2016)* and puts it into the top 5% of transcriptome assemblies by the overall assembly quality score. On the individual level, 51,990 out of 60,180 transcripts, or 86%, are classified by TransRate as 'good' (*Table 1*, *Figure 1—figure supplement 2A*, *Figure 1—source data 1*). The remaining 8190 transcripts might have assembly errors, but we decided to keep them in the assembly, since some genuine low-expressed transcripts might fall into this category. TransRate contig scores (*Figure 1—source data 1*) are included in the transcriptome annotation to facilitate transcript filtering as needed.

The assembly appears to be complete in terms of genes space, with 247 out of 248 core eukaryotic genes (*Parra et al., 2009*) present (*Table 1*). Benchmarking Universal Single-Copy Orthologs (BUSCO) assessment of the transcriptome (*Simão et al., 2015*) using 303 Euakaryotic dataset genes reveals 296 complete, one fragmented and six missing gene models (*Figure 1—figure supplement 2B*). This BUSCO distribution is very similar to the assessment of the *Schmidtea mediterranea* transcriptome assembly Smed_dd_v6 (*Figure 1—figure supplement 2B*), which is commonly used in the planarian field (*Liu et al., 2013*; *Wurtzel et al., 2015*; *Solana et al., 2016*). However, in contrast to the *S. mediterranea* transcriptome assembly, more than half of the complete gene models are not single-copy but duplicated in the *M. lignano* transcriptome (*Figure 1—figure supplement 2B*). The presence of multiple copies of the genes that usually are single-copy in other organisms can be explained by the observation that *M. lignano* DV1 line used for the transcriptome assembly has a duplicated large chromosome, and hence a likely recent partial genome duplication (*Zadesenets et al., 2016*).

Furthermore, MLRNA150904 transcriptome assembly has 3503 different PFAM domain annotations, 8458 identifiable homologs of human genes, and 1697 homologs of *S. mediterranea* cell-type-specific genes (*Wurtzel et al., 2015*). More than 10% of the transcripts appeared to be trans-spliced (*Table 1*).

Since the alternatively spliced transcripts in the de novo assembly can be difficult to assign correctly to the genes, we found it helpful in gene expression studies to use the Corset tool (*Davidson and Oshlack, 2014*), which performs hierarchical clustering of transcripts based on mapped reads and generates clusters of transcripts (a proxy to genes) and gene-level counts.

## Transcriptome of proliferating cells: irradiation approach

Worms were irradiated with three doses of 70 Gy within 1 day. As this protocol differs from the previously published approach (*De Mulder et al., 2010*), we re-examined morphology, survival, mitotic activity, and gene expression after irradiation to confirm the elimination of all proliferating cells.

At the morphological level, irradiation induced several changes. Immediately after the third irradiation pulse, gonads could not be observed. Other defects appeared after 14 days post irradiation: worms shrunk, deformations such as blisters and bulges appeared, and eventually worms disintegrated into pieces (*Figure 1—figure supplement 3A*). From 14 days after irradiation, survival decreased, with 100% mortality reached after 35 days (*Figure 1—figure supplement 3B*). The effect of γ-irradiation on the number of mitotic cells was examined at three time points. At 12 and 24 hr post irradiation, no mitotic activity was detected. At 72 hr, a few labeled cells were observed (*Figure 1—figure supplement 3C*).

To establish at which time point the proliferating cells are eliminated, we determined which genes have a significant diminished expression between 0 hr and 12 hr, 12 hr and 24 hr, and between 24 hr and 72 hr after irradiation (*Figure 1—source data 2*). The largest effect was observed at 12 hr post-irradiation, with 8929 downregulated transcript clusters (FDR < 0.05), of which 3548 were downregulated by more than twofold (*Figure 2A*). Substantially smaller changes were observed at subsequent time points, with 3870 and 1732 downregulated transcript clusters between 12 hr and 24 hr and 24 hr and 72 hr, respectively (*Figure 2A*). GO term analysis of the transcripts depleted at the 12 hr time point revealed enrichment for several processes known to be associated with neoblast genes (*Rossi et al., 2007*; *Rink, 2013*), such as RNA metabolism, DNA replication, cell cycle, and chromatin modification (*Figure 1—source data 3*). Notably, this GO-term enrichment was not present in the later time points. Moreover, analysis of the distribution of homologs of *S. mediterranea* cell type markers from *Wurtzel et al. (2015)* revealed 105 out of 157 'Neoblast' markers among transcripts

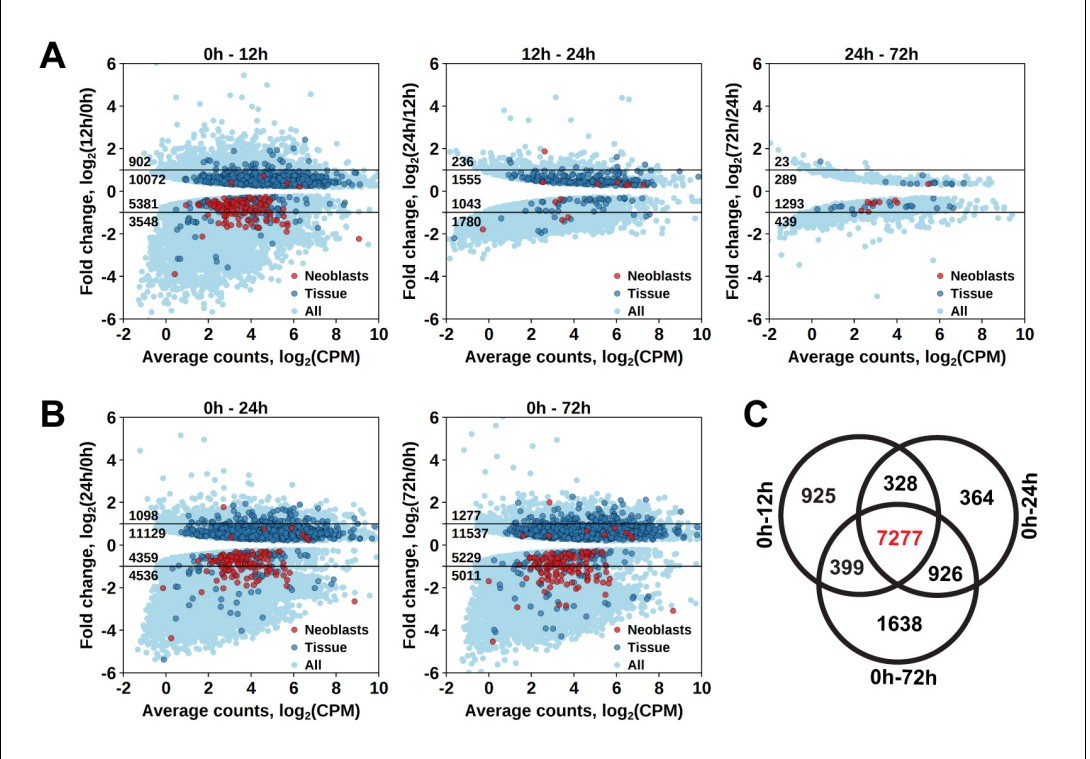

**Figure 2.** Identification of differentially expressed genes based on γ-irradiation approach. (**A**) Temporal profile of differentially expressed genes between all three time points. (**B**) Genes differentially expressed between 0 hr and 24 hr and 0 hr and 72 hr. Classification of genes as 'Neoblast' and 'Tissue' in **A** and **B** is based on homology to *S. mediterranea* genes from *Wurtzel et al. (2015)*. (**C**) Venn diagram representation of the number of genes enriched in proliferating cells (indicated in red).

depleted at the 12 hr time point (*Figure 2A*), which is a 3.3-fold enrichment relative to the expected from random distribution ($p<10^{-15}$, Pearson's Chi-squared test with Yates' continuity correction). 'Neoblast' was the only cell type enriched among down-regulated transcripts at the 12 hr time point, and no further enrichments were observed at later time points (*Figure 1—source data 4*). Taken together, the data shows that three pulses of 70 Gy within 1 day is a lethal dose killing the proliferating cells, including neoblasts, within 12 hr after irradiation.

As proliferating cells are killed between 0 hr and 12 hr after irradiation, genes specifically expressed in them should be permanently downregulated from 12 hr post-irradiation onwards. Indeed, the number of significantly downregulated genes between 0 hr and 12 hr, and 0 hr and 24 hr, and 0 hr and 72 hr was similar, with an increase at the 72 hr time point (*Figure 2A,B*), and the genes largely overlapped, with 7277 transcript clusters downregulated at all three time points (*Figure 2C* and *Figure 1—source data 2*). Similarly, the only significantly enriched *Schmidtea* cell type in this dataset was 'Neoblast' (3.78-fold enrichment, $p<10^{-15}$) (*Figure 1—source data 4*), and the enriched GO-terms included nucleic acid metabolic processes, cell cycle, DNA replication and chromosome organization (*Figure 1—source data 3*). While this gene set is characteristic for proliferating cells in *M. lignano*, it does not allow distinguishing between proliferating neoblasts and germ line cells, and therefore, another approach was required for this purpose.

## Transcriptome of proliferating cells: FACS approach

A multistep gating strategy based on a live cell Hoechst staining was developed to sort a population of differentiated cells with a 2C DNA content, and a population of proliferating cells in late S, G2, and M-phases with a 4C DNA content (*Figure 1—figure supplement 4A*). Irradiation of animals before sorting resulted in a six-fold decrease of the fraction of cells in the 4C gate (*Figure 1—figure*

*supplement 4B*), confirming that this gate represents proliferating cells and not contamination of e.g. doublets of differentiated cells.

By comparing gene expression of the 2C and 4C cell populations of intact adult worms, we established a list of 7124 transcript clusters significantly enriched in proliferating cells (S/G2/M cells), of which 5264 were upregulated by more than twofold (*Figure 3A* and *Figure 1—source data 2*). Of those, 3374 transcript clusters were also identified with the irradiation method. Similar to the irradiation experiments, the enriched GO terms in this list of genes include nucleic acid metabolic process, RNA processing, DNA replication, chromosome organization, and cell cycle processes (*Figure 1—source data 3*). Moreover, the *Schmidtea* 'Neoblast' cell type is again the only significantly enriched

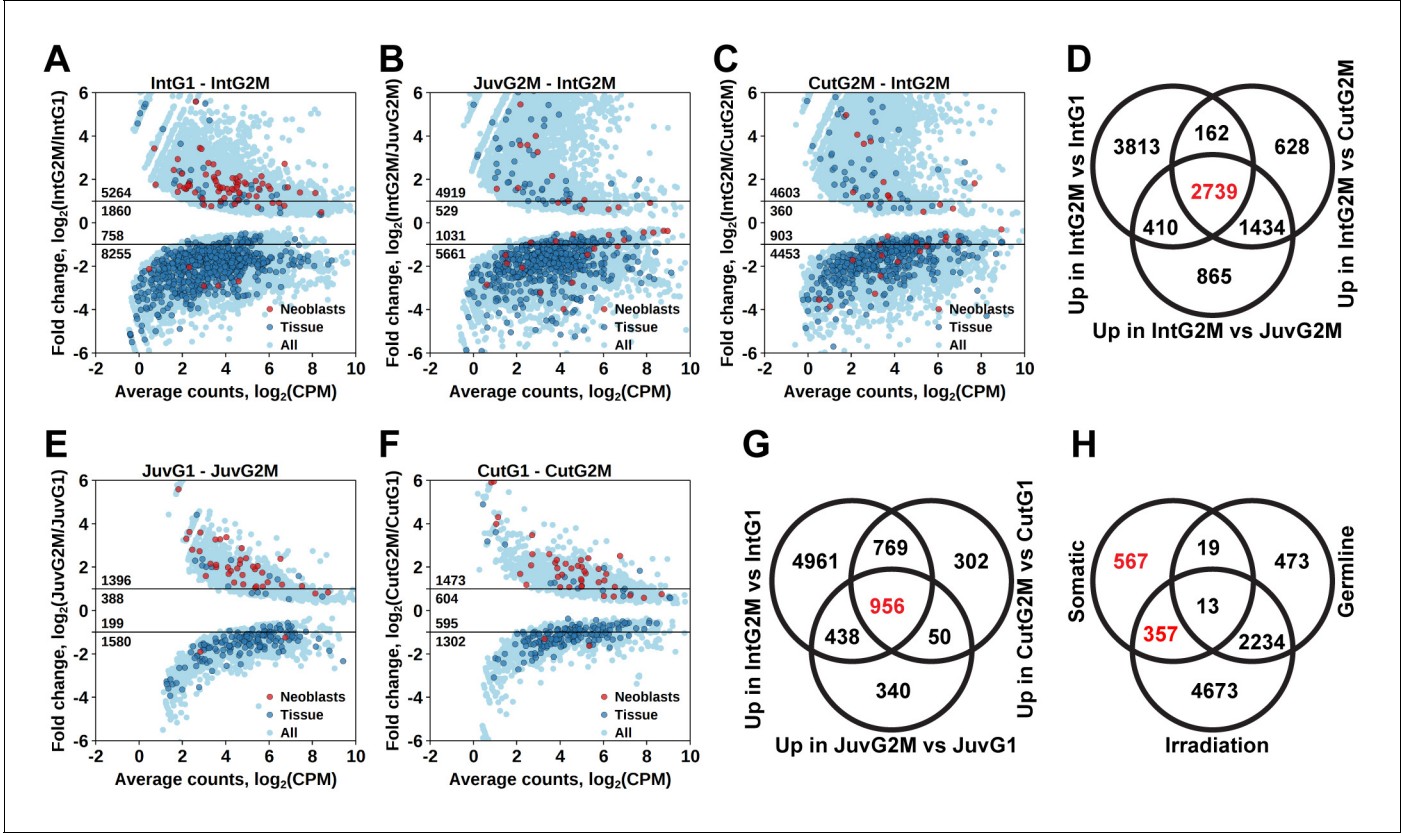

**Figure 3.** Identification of differentially expressed genes based on FACS approach. (A) Genes differentially expressed between differentiated (G1 phase of cell cycle, 2C DNA content) and proliferating (G2/M phase of cell cycle, 4C DNA content) cells of intact worms. (B) Genes differentially expressed between proliferating cells of juvenile and intact worms. (C) Genes differentially expressed between proliferating cells of cut and intact worms. (D) Venn diagram representation of the number of genes enriched in the germline (indicated in red). (E) Genes differentially expressed between differentiated and proliferating cells of juvenile worms. (F) Genes differentially expressed between differentiated and proliferating cells of cut worms. (G) Venn diagram representation of the number of genes enriched in somatic neoblasts (indicated in red). (H) Venn diagram representation of the number of genes enriched in somatic neoblasts based on both approaches: irradiation and FACS (indicated in red). Classification of genes as 'Neoblast' and 'Tissue' in A, B, C, E, and F is based on homology to *S. mediterranea* genes from *Wurtzel et al. (2015)*.

The following source data and figure supplements are available for figure 3:

**Source data 1.** Germline candidate genes, in situ hybridization and RNAi results.

**Source data 2.** Stringent neoblast candidate genes and RNAi results.

**Figure supplement 1.** Gene expression patterns for genes enriched in the germline.

**Figure supplement 2.** *Mlig-cpeb1* and *Mlig-ddx6* RNAi-phenotypes and ISH.

type in S/G2/M cells (2.68-fold enrichment, $p<10^{-15}$), whereas other tissue markers are enriched among transcripts specific for the G1 cell population (*Figure 3A* and *Figure 1—source data 4*).

## Distinguishing proliferating cells in the soma and germline

As *M. lignano* is a hermaphrodite, its proliferating cells include both germline (stem) cells and somatic neoblasts. For further studies and development of markers, it is important to determine which genes are enriched in the proliferating cells of the germline or of the soma. For this purpose, genes expressed in late-S/G2/M cells of intact worms (*Figure 3A*) were compared to genes expressed in late-S/G2/M cells of hatchlings (*Figure 3B*) and amputated heads (*Figure 3C*). Since the latter two conditions do not contain gonads, this allows distinguishing proliferating cells of germline and soma, and 2739 transcript clusters enriched in the germline were identified (*Figure 3D* and *Figure 1—source data 2*). Only 492 of those transcript clusters do not have a down-regulated expression after irradiation (*Figure 3H*). The list includes known germline-specific genes, such as *macboule, melav* (*Sekii et al., 2009*; *Kuales et al., 2011*), and several genes identified in the positional RNA-Seq dataset of *M. lignano* (*Arbore et al., 2015*) as gonad-specific (e.g. RNA815_7008, RNA815_9973.1, RNA815_1618.1, RNA815_2640, RNA815_7725.2, RNA815_12337.1). Investigation of expression patterns of 27 candidate genes by in situ hybridization confirmed in all cases their expression in gonads, either in testes or in both testes and ovaries (*Figure 3—figure supplement 1*, *Figure 3–figure supplement 2* and *Figure 3—source data 1*). We knocked down 17 of these gonad genes by RNAi (*Figure 3—source data 1*), and screened for obvious changes in gonad morphology within 3 weeks. This resulted in phenotypes for two genes: *Mlig-cpeb1* and *Mlig-ddx6*. In case of *Mlig-cpeb1*, the testes were enlarged, ovaries often became less distinct, and developing eggs were absent. Amputated tails could regenerate. In case of *Mlig-ddx6*, both testes and ovaries disappeared, and worms obtained a wrinkled appearance. When tails were amputated at the eighth day of treatment, no blastema was formed, resulting in the lack of regeneration (*Figure 3—figure supplement 2*).

To elucidate genes enriched in somatic neoblasts, transcripts enriched in sorted proliferating cells of intact worms (*Figure 3A*), hatchlings (*Figure 3E*), and amputated heads (*Figure 3F*) were overlapped, resulting in 956 transcript clusters (*Figure 3G*). We further filtered this list by excluding germline enriched transcripts (*Figure 3D*), even though the overlap was minimal and 924 out of 956 transcript clusters remained and were classified as enriched in somatic neoblasts (*Figure 3H* and *Figure 1—source data 2*) Indeed, this list contains 26 *S. mediterranea* 'Neoblast' marker homologs out of the 157 annotated in the transcriptome (*Figure 1—source data 2*), which is a 7.9-fold enrichment ($p<10^{-15}$) relative to the random distribution (*Figure 1—source data 4*).

To further narrow the list of genes enriched in somatic neoblasts, we overlapped transcript clusters identified as somatic neoblasts by the FACS approach (*Figure 3H*) with the irradiation-derived list of transcript clusters enriched in proliferating cells (*Figure 2C*). This resulted in 357 transcript clusters enriched in somatic neoblasts (*Figure 3H*), which we termed Stringent Neoblast candidate genes (*Figure 1—source data 2* and *Figure 3—source data 2*). The previously mentioned GO term enrichments characteristic of neoblast remained in this narrowed down list (*Figure 1—source data 1*), and, the *Schmidtea* 'Neoblast' cell type is highly enriched and contains 22 out of the 157 'Neoblast'-annotated transcripts (17.3-fold enrichment, $p<10^{-15}$, *Figure 1—source data 4*). Furthermore, 211 out of the 357 Stringent Neoblast transcript clusters have identifiable human homologs representing 159 different human genes (*Figure 1—source data 2*).

## Identifying genes essential for somatic neoblast functionality in *Macrostomum lignano*

To validate the generated list of Stringent Neoblast candidate genes and investigate their role in neoblast biology, we performed RNAi knockdown experiments on a selected set of transcripts with identifiable human homolog genes (*Figure 3—source data 2*). These included genes known to be involved in cell cycle processes, as well as candidates that were previously not studied in the context of stem cell functionality and regeneration. During the screen, we focused on genes giving fast and robust phenotypes. In other words, phenotypes, which were observed within 3 weeks, in all treated worms, based on morphology. Out of the 14 genes tested, 5 gave the fast phenotype: *Mlig-ddx39*, *Mlig-rrm1*, *Mlig-rpa3*, *Mlig-cdk1*, and *Mlig-h2a* (*Figure 4A* and *Figure 3—source data 2*). In

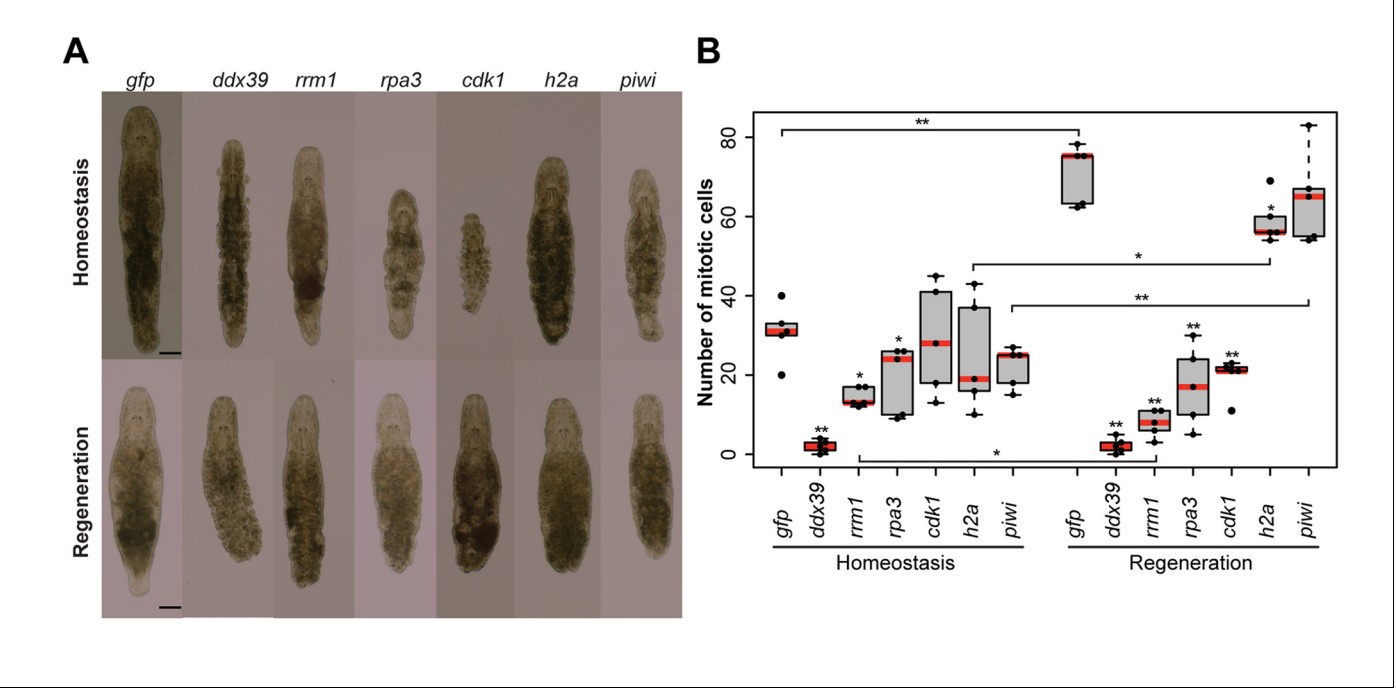

**Figure 4.** RNA interference screen. (**A**) Effects of gene knockdown on homeostasis and regeneration capacity. Phenotypes often include: shrinkage of the worms, appearance of bulges, disappearance of the gonads, and the lack of regeneration after amputation of the tail. Scale bar 100 μm. (**B**) Effects of RNAi on the number of mitotic cells during homeostasis and regeneration. Each dot represents one animal. In the Homeostasis group, stars represent significant differences compared to *gfp(RNAi)* homeostasis animals. In the Regeneration group, stars represent significant differences compared to *gfp(RNAi)* regenerating animals. Lines represent significant differences between cut ant intact worms. *p<0.05, **p<0.001 (two sample *t*-test).

The following figure supplement is available for figure 4:

**Figure supplement 1.** *Mlig-pcna* and *Mlig-cdc20* RNAi-phenotypes.

addition, the previously described phenotype for *Mlig-piwi* (*De Mulder et al., 2009*) was confirmed, and two less severe phenotypes were found: *Mlig-pcna* and *Mlig-cdc20* (*Figure 4—figure supplement 1*).

Homologues of ribonucleotide reductase M1 (*RRM1*), replication protein A3 (*RPA3*), cyclin-dependent kinase 1 (*CDK1*), proliferating cell nuclear antigen (*PCNA*), and cell division cycle protein 20 (*CDC20*) are encoding proteins previously linked to cell cycle processes (*Lee and Nurse, 1987*; *Travali et al., 1989*; *Henricksen et al., 1994*; *Parker et al., 1995*; *Weinstein, 1997*). Histone family member A (*H2A*) was shown to be important for packaging DNA into chromatin and is consequently involved in gene expression regulation (*Mariño-Ramírez et al., 2006*). To our knowledge, DEAD box polypeptide 39 (*DDX39*) was not previously studied in the context of in vivo stem cell functionality. All five robust phenotypes suggest problems with cell turnover during both homeostasis and regeneration. During homeostasis, the body cannot be maintained as the gonads disintegrate, worms often shrink, and wrinkles and bulges appear. After amputation of the tail, the regeneration capacity is completely lost as a blastema cannot be formed. As a result, the wound is closed but amputated structures are not regenerated. In case of the *DDX39* homologue, all treated worms died within 3 weeks. For the other genes, most worms were still alive after 3 weeks of RNAi-treatment. Knocking down the gene encoding piwi like-1 protein (*PIWI1*), which was previously used as a neoblast marker in *M. lignano*, resulted in a similar phenotype (*Figure 4A*). The regeneration phenotype is, however, less severe than that of the other five genes, as a small blastema can be observed in several worms. This blastema stays small and does not differentiate, resulting in the lack of regeneration of the lost body parts. The *Mlig-pcna* phenotype is characterized by regeneration after amputation of a small ventrally oriented tail. During homeostasis, gonads disintegrate and a wrinkled

appearance of the body can be observed. An additional treatment, during which the tail was amputated after 3 weeks of treatments, resulted in the complete lack of regeneration. The *Mlig-cdc20 (RNAi)* phenotype is limited within the first 3 weeks of treatment as the only effect is the disintegration of gonads, and defects in regeneration are not observed (*Figure 4—figure supplement 1*).

To investigate whether the five robust and *Mlig-piwi* phenotypes can be related to changes in the proliferation rate of neoblasts, the number of mitotic cells was determined at the 10th day of RNAi in both cut and intact worms (*Figure 4B*). Compared to the *gfp(RNAi)* control intact animals, the number of mitotic cells in intact worms is significantly decreased during RNAi of *Mlig-ddx39* (p<0.001, *t*-test), *Mlig-rrm1* (p=0.001, *t*-test), and *Mlig-rpa3* (p=0.048, *t*-test). Amputation of the tail increased the effect of knockdown on the number of mitotic cells. Compared to the *gfp(RNAi)* control cut worms, there are significantly less mitotic cells for RNAi of *Mlig-ddx39* (p<0.001, *t*-test), *Mlig-rrm1* (p<0.001, *t*-test), *Mlig-rpa3* (p<0.001, *t*-test), *Mlig-cdk1* (p<0.001, *t*-test), and *Mlig-h2a* (p=0.027). In addition, it is interesting to compare the amount of mitotic cells between intact and cut worms as it has been shown that amputation of the tail results in a significant increase of mitotic cells 48 hr after amputation (*Nimeth et al., 2002*). In case of *Mlig-ddx39*, less than five mitotic cells per worm are observed in both intact and cut individuals. Consequently, the number of mitotic cells does not increase after tail-amputation (p=0.862, *t*-test). RNAi of *Mlig-rrm1* even results in a significant decrease of mitotic cells after tail amputation (p=0.008, *t*-test). A non-significant decrease after tail amputation is observed during knockdown of *Mlig-rpa3* (p=0.771, *t*-test) and *Mlig-cdk1* (p=0.112, *t*-test). In the case of *Mlig-h2a(RNAi)* and *Mlig-piwi(RNAi)*, amputation of the tail still results in a significantly increased amount of mitotic neoblasts (*Mlig-h2a*: p=0.001, *t*-test; *Mlig-piwi*: p<0.001, *t*-test).

## DDX39 as a novel marker for proliferating cells

Knockdown of *Mlig-ddx39* gene proved to be the most severe phenotype of all screened candidates. In situ hybridization performed on adults and juveniles revealed the typical expression pattern of a gene enriched in proliferating cells (*Figure 5A*), known from the published expression of *piwi* (*De Mulder et al., 2009*). Both genes are expressed in the testes, ovaries, developing eggs and the somatic neoblasts, visualized by bilateral bands (*Figure 5A,B*). In situ hybridization 12 hr post-amputation confirmed the expression of *Mlig-ddx39* and *Mlig-piwi1* in the blastema region, which consists of proliferating neoblasts (*Figure 5C,D*). An antibody against *M. lignano* PIWI1 protein was previously developed and demonstrated to label neoblast population (*Wasik et al., 2015*). A combined *Mlig-ddx39* FISH / Macpiwi1 antibody labeling revealed cells co-expressing *Mlig-ddx39* and *Macpiwi1* in the testes, ovaries and somatic neoblasts (*Figure 5E–G*). Furthermore, a combined *Mlig-ddx39* FISH/phospho-histone H3 antibody mitotic labeling revealed expression of *Mlig-ddx39* in the proliferating cells of blastema (*Figure 5H*). These observations provide additional evidence for neoblast-specific expression of *Mlig-ddx39*.

## Conservation of stem cell genes between *Macrostomum*, planarians, and mammals

The generated *Macrostomum* gene sets enriched in proliferating cells and in germline allow probing the evolutionary conservation of the involved genetic pathways within flatworms and beyond. Toward this end, we overlapped *M. lignano* neoblast and germline transcript categories with transcripts related to the neoblasts of *S. mediterranea* (*Onal et al., 2012*; *Wurtzel et al., 2015*), the germline or sexual strain of *Schmidtea* (*Wang et al., 2010*; *Chong et al., 2011*; *Resch et al., 2012*), and to mammalian pluripotency genes (*Tang et al., 2010*).

Many of the *M. lignano* transcripts identified as enriched in proliferating cells by the irradiation approach have homologs within the set of X1 neoblast transcripts in *Schmidtea* (*Figure 1—figure supplement 5A,C*), and the overlap is 3.78-fold higher than expected from random distribution (*Figure 1—source data 4*), suggesting the presence of the subset of genes with conserved neoblast function in flatworms. Furthermore, comparison with mammalian genes known to be involved in pluripotency (*Tang et al., 2010*) shows results similar to those previously reported for *Schmidtea* (*Onal et al., 2012*), with a substantial overlap between *Macrostomum* neoblast and irradiation categories and mammalian pluripotency maintenance genes (*Figure 1—figure supplement 5D*), which are 4.08-fold overrepresented in the stringent neoblast category (*Figure 1—source data 4*). At the

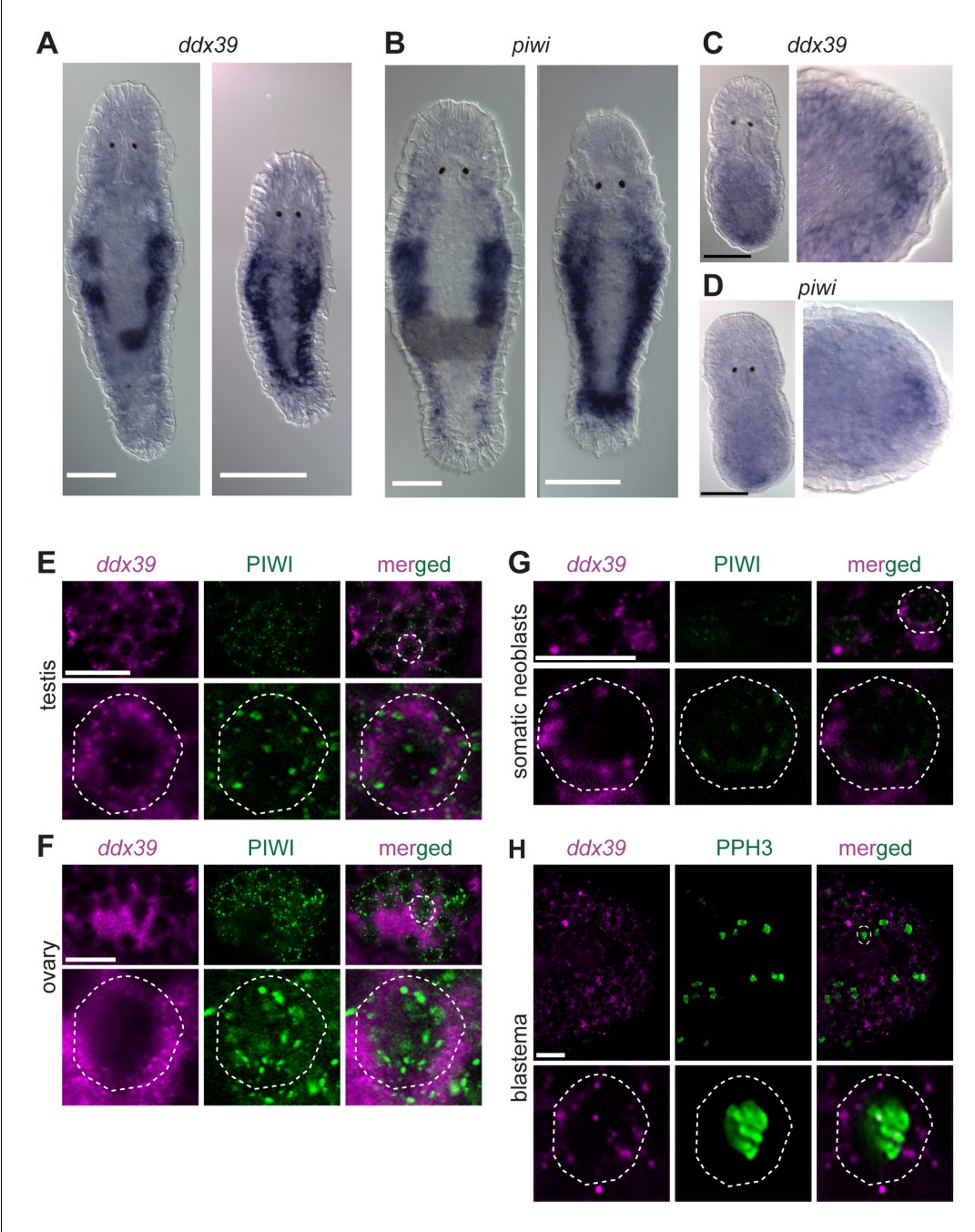

**Figure 5.** *Mlig-piwi* and *mlig-ddx39* expression patterns. (A,B) WISH expression pattern in adult and juvenile worms for *Mlig-ddx39* and *Mlig-piwi*. Both genes are expressed in the testes, ovaries, developing eggs and somatic stem cells located in bilateral bands. (C,D) WISH expression patterns for *Mlig-ddx39* and *Mlig-piwi* in the blastema, 12 hr post-amputation. (E–G). *Mlig-ddx39* FISH and Macpiwi antibody double labeling of testis, ovary, and somatic neoblasts. (H) *Mlig-ddx39* FISH and mitotic phospho H3 antibody double labeling of the wound site, 48 hr post-amputation. Individual cells are outlined and magnified in the second rows in panels *E-H*. Scale bars: 100 µm (A–D) and 25 µm (E–H).

same time, pluripotency repressor genes are nearly completely excluded from the neoblast category (*Figure 1—source data 4*).

The transcripts which are enriched in the proliferating germline cells of *M. lignano* only show a small overlap with known regulators of germ cell development in *S. mediterranea* and genes which are enriched in or specific for the sexual strain compared to the asexual strain of *S. mediterranea*

(*Figure 1—figure supplement 5D*). The planarian germline set is based primarily on the differentiated cells, while the *Macrostomum* set is enriched for genes expressed in proliferating germline cells, which explains the minimal overlap between these sets.

## Discussion

Obtaining insight in the gene expression profile of somatic neoblasts of *M. lignano* was the primary aim of this study. This included several steps: determining a de novo transcriptome of *M. lignano*, identifying genes enriched in its proliferating cells, and distinguishing which of those genes are enriched in the germline or in the somatic neoblasts. As a last step, RNAi was used to confirm the essential role of five genes for the functionality of somatic neoblasts.

A dual approach was used to determine which genes are enriched in proliferating cells: irradiation and FACS. Both techniques proved to be efficient in determining the gene expression profile of planarian neoblasts (*Reddien and Sánchez Alvarado, 2004*; *Hayashi et al., 2006*; *Rossi et al., 2007*; *Eisenhoffer et al., 2008*; *Blythe et al., 2010*; *Galloni, 2012*; *Shibata et al., 2012*; *Solana et al., 2012*) and were, therefore, our first choice.

In this study, worms were irradiated with three doses of 70 Gy within 1 day to be able to study changes in gene expression as a function of time after irradiation. This differs from the published lethal fractionated protocol performing several doses distributed over 9 days and accumulating in 210 Gy (*De Mulder et al., 2010*). However, the results of both methods on the level of morphology, mortality, and decreasing number of proliferating cells are very similar. This demonstrates that the previously described fractionation of doses can be given within 1 day without decreasing the efficiency of the treatment. The effect of irradiation on gene expression is fast and mainly takes place within the first 12 hr after the last irradiation pulse (*Figure 2A*). It was shown before that irradiation causes a broad stress reaction to the cells, triggered by global DNA damage, but also includes changes due to neoblast elimination (*Eisenhoffer et al., 2008*; *Solana et al., 2012*). In *M. lignano*, analysis of GO Terms and *Schmidtea* cell type enrichment demonstrates that genes downregulated at 12 hr post-irradiation are enriched for genes specific for proliferating cells. As the elimination of proliferating cells should be seen at the RNA-seq level as a permanent downregulation of genes specifically expressed in these cells, only transcript clusters that are downregulated in all the three studied time points were selected. In this way, false positives with a temporal decreased expression were avoided.

The *M. lignano* FACS strategy was developed to isolate populations of differentiated and proliferating cells. It focuses on live cell Hoechst labeling of the nuclear content, as cells in the late S, G2, and M phase of the cell cycle have double the amount of DNA (4C) than differentiated cells (2C) (*Figure 1—figure supplement 3*). Hoechst labeling has been commonly used to isolate proliferating cells in planarians (*Hayashi et al., 2006*; *Higuchi et al., 2007*; *Eisenhoffer et al., 2008*; *Shibata et al., 2012*), and the 4C DNA content is one of the main characteristics of the X1 cells, which are highly enriched in neoblasts (*Hayashi et al., 2006*). Before the selection of cells based on nuclear content, a few additional gating steps are performed to remove cell debris and cell clusters from the selection, but also to be able to identify the 4C population. This population is much smaller than the 2C population, as 'only' 6.5% of all cells in *M. lignano* are neoblasts (*Bode et al., 2006*). As all approaches, the FACS strategy has its own limitations, which is mainly the focus on actively dividing cells. Consequently, genes specific for the small population of quiescent neoblasts (*Verdoodt et al., 2012*) will be missed.

Both approaches resulted in a set of about 7000 transcript clusters, clearly representing genes enriched in proliferating cells based on the *Schmidtea*-'Neoblast' cell type enrichment and GO Term analysis. While both RNA-Seq datasets are largely enriched for the same GO Terms, there are still differences indicating the bias of the two approaches. The irradiation dataset, for example, is more enriched for processes related to cell cycle and DNA repair, while the gene set identified by FACS has a higher enrichment for processes related to RNA processing (*Figure 1—source data 3*). The bias is further shown by the fact that about half of the identified transcript clusters are specific for the used method. The other half (3374 transcript clusters) are identified with both approaches and represent a trustworthy selection of genes enriched in proliferating cells. Detailed information can be found in *Figure 1—source data 2*.

As *M. lignano* is a sexually reproducing flatworm with testes and ovaries, it is important to distinguish which of the identified genes are enriched in germline cells and which in somatic neoblasts. Therefore, differentiated and proliferating cells were not only isolated from adults, but also from hatchlings and amputated heads, both lacking gonads. This approach proved to be efficient, as only 32 transcript clusters are overlapping in the identified sets of genes enriched in germline cells and in somatic neoblasts (*Figure 3*).

Interestingly, much more transcript clusters are enriched in the germline (2739) than in the somatic neoblasts (956). Previous publications (*Sekii et al., 2009*; *Kuales et al., 2011*; *Arbore et al., 2015*), and a first ISH-screen of a selection of transcripts enriched in germline cells identified in this study, confirmed that several of these genes are enriched or even expressed specifically in the gonads. The essential function for the gonads was confirmed in the study for two genes: *Mlig-cpeb1* and *Mlig-ddx6*. Knockdown of *Mlig-ddx6*, however, also resulted in regenerative defects, indicating that despite the enrichment of the gene in gonads, it still has a crucial role in somatic neoblast functionality. In situ hybridization experiments show that the majority of genes enriched in the germline seem to be specific for the testes (*Figure 3—source data 1*). It has been suggested that the large amount of testes-specific genes reflects the functional complexity of the testes and the requirement of producing highly elaborate sperm (*Arbore et al., 2015*).

Several previous studies in planaria revealed sets of genes required for germ cell development (*Wang et al., 2010*; *Chong et al., 2011*) or enriched in sexual *S. mediterranea* animals (*Resch et al., 2012*). However, the overlap between these genes and *M. lignano* germline gene set identified in this study is minimal (*Figure 1—figure supplement 5B*). When considering genes regulating the development, regeneration, and maintenance of gonads and gametes, it is important to distinguish between the proliferating and differentiated germline cells. This can explain the small overlap between the transcripts enriched in the proliferating germline cells of *M. lignano* and the published datasets of *Schmidtea* germline genes, which primarily include transcripts specific for differentiated germline cells. Establishment of *M. lignano* gene sets specific for differentiated gonad cells will be required for future comparative studies of germ cell development in flatworms.

In situ hybridization (ISH) of transcripts enriched in somatic neoblasts showed that, despite the enrichment, these genes are also expressed in the gonads. Due to limitations of the ISH technique to visualize single neoblasts scattered in the mesenchyme, expression in the gonads is more obvious than expression in the somatic neoblasts (*Figure 5*). Therefore, it is important to develop more sensitive methods for visualizing gene expression in single cells in *M. lignano*, as this will be essential to confirm the specificity of genes for the somatic neoblasts or the germline. Our first attempts on implementing fluorescent in situ hybridization in *M. lignano* based on protocols developed for planarians (*Currie et al., 2016*) are encouraging (*Figure 5E–H*), but it remains to be demonstrated how robust the method is when applied to a larger selection of genes.

The current lack of genes specifically expressed in somatic neoblasts of *M. lignano* is fascinating, and it is unclear whether they do not exist and neoblasts residing in mesenchyme and in gonads are largely similar, or cannot be identified yet due to technical limitations. In the future, it will be important to identify transcriptional signatures of somatic neoblasts by using a combination of markers enriched in somatic neoblasts and germline-specific markers, which should be absent in these cells. In addition, it would be interesting to identify the differentiation lineage from somatic stem cell to gamete as *Macrostomum* can regenerate its gonads. Gene expression data obtained in this study can facilitate the design of such experiments.

To obtain a final selection of genes enriched in somatic neoblasts, and with a potential crucial role for their functionality, we combined the irradiation and FACS data, resulting in 357 transcript clusters. Among these transcript clusters, 211 have clear homology to 159 different human genes, and therefore are particularly interesting to study evolutionary conserved aspects of stem cell functioning. The observation that mammalian homologs with a known function in pluripotency maintenance are enriched more than four fold in the stringent neoblast set (*Figure 1–source data 4* and *Figure 1—figure supplement 5B*) further illustrates this. The remaining 146 genes are not conserved in humans and could be interesting in explaining the astonishing regeneration capacity of flatworms. Moreover, such candidates could be studied in the context of parasitic flatworms, as they are potential therapeutic targets.

To explore the importance of the first selection of genes for neoblast functionality, we performed RNAi experiments studying both homeostasis and tail regeneration. By using 3-weeks screens, five

genes resulting in clear phenotypes were found (*Mlig-ddx39*, *Mlig-rrm1*, *Mlig-rpa3*, *Mlig-cdk1*, and *Mlig-h2a*). In addition, *Mlig-piwi*, the only known gene essential for neoblast functionality in *M. lignano* (*De Mulder et al., 2010*), was identified again, and two less severe phenotypes were observed within the 3-weeks screens (*Mlig-pcna* and *Mlig-cdc20*). Most of these genes are known to be involved in cell cycle regulation (*Lee and Nurse, 1987*; *Henricksen et al., 1994*; *Parker et al., 1995*). In addition, genes encoding histone proteins were previously shown to be important for neoblasts in *S. mediterranea*, as *Smed-H2B* is essential for neoblast maintenance (*Solana et al., 2012*). The morphological changes of the five robust phenotypes during homeostasis are very similar to those of irradiation and suggest problems with cellular turnover and stem cell survival. In addition, no blastema could be observed after tail-amputation, and mitotic labeling revealed a significant decrease in proliferation, confirming this hypothesis. In the case of the *Mlig-piwi* and *Mlig-pcna* phenotypes, a small blastema and even a small tail, respectively, could be formed (*Figure 4—figure supplement 1*). Amputation of the tail after 3 weeks of treatment with *Mlig-pcna* dsRNA resulted in a complete lack of a blastema, indicating that *Mlig-pcna* needs to be knocked down for a longer period before effects can be observed. As disintegration of gonads is often the first sign of a phenotype during homeostasis, more research is needed to study whether the *Mlig-cdc20* phenotype is much slower than the others and develops with further treatment, or is limited to the germline. It is important to note that longer and more detailed RNAi screens might reveal additional slower or subtler phenotypes, and it is important to design RNAi screens according to the scientific question. For this study, we chose to focus on fast, obvious, and robust phenotypes visible in all worms within 3 weeks, as we were mainly interested in identifying genes that can be used as neoblast markers and experimental controls for further more detailed research of stem cell biology in *M. lignano*.

In our RNAi screen, *Mlig-ddx39* stood out as the fastest and most severe phenotype, and therefore clearly essential for neoblast functionality. Knockdown of *Mlig-ddx39* resulted in the death of all worms within 3 weeks, which was not the case with other tested candidate genes. The mitotic labeling revealed that already during the 10[th] day of treatment almost no mitotic cells can be observed in both cut and intact *Mlig-ddx39(RNAi)* animals, suggesting the fast elimination of proliferating cells. This explains the morphological changes, such as shrinkage, bulges, loss of gonads, and general disintegration, which already become visible during the second week of treatment. Based on these results, we propose the use of *Mlig-ddx39* as a positive control for RNAi experiments in *M. lignano*, which is already a common practice in our laboratory. In addition, the ISH results showing enriched expression in the gonads and in somatic neoblasts in both the intact body and in the blastema indicate that *Mlig-ddx39* could be a convenient marker for proliferating cells.

Interestingly, *ddx39* is highly conserved in different species, as all members of the DEAD box RNA helicase family. These genes include a common D-E-A-D (Asp-Glu-Ala-Asp) motive and are known for their roles in RNA metabolism (*Linder and Jankowsky, 2011*). Functions of *ddx39* have been linked to mRNA export, which was demonstrated in *D. melanogaster* (*Eberl et al., 1997*) and *C. elegans* (*MacMorris et al., 2003*). Moreover, *ddx39* was shown to be important for regeneration and development of limbs in *X. laevis* (*Wilson et al., 2010*). Therefore, it could be interesting to use different model organisms for studying whether *ddx39* has a conserved function in stem cell biology.

To increase the accessibility of the generated datasets, we developed an online interface to this resource, which is available at http://neoblast.macgenome.org. The interface provides a straightforward way to search through the different transcript categories and to visualize and analyze the expression data of any gene of interest, for example by transcript ID, gene name or keyword. In addition, links to download the transcriptome assembly and the gene expression data and the classification into categories are provided. To facilitate the comparison of *M. lignano* and planarians, *S. mediterranea* homologs and their various classifications are provided, as well as links to PlanMine, which contains comprehensive information of planarian genomics (*Brandl et al., 2016*).

In summary, the de novo *M. lignano* transcriptome and generated sets of genes enriched in the germline and somatic neoblasts are valuable resources for further development of this species as a model organism for stem cell research. A preliminary screen already identified several novel, not previously implicated in stem cell biology, genes essential for neoblast functionality. Specifically, *ddx39* is suggested as a positive control for RNAi experiments and a marker for proliferating cells.

## Materials and methods

### Culture of *Macrostomum lignano*

*Macrostomum lignano* is cultured in Petri dishes with nutrient-enriched artificial seawater (f/2) (*Anderson et al., 2005*), at 20°C and a 14 hr/10 hr light/dark cycle. Worms are fed *ad libitum* with the diatom *Nitzschia curvilineata* (*Rieger et al., 1988*).

### γ- Irradiation treatment

Worms were exposed to γ-rays of 0,0288 cGy/sec with a Cesium 137 γ-ray machine (CIS Bio International S.A, France) at the Department of Cellular Biology, University Medical Center Groningen. Batches of animals (n = 120) kept in a Petri dish with f/2 were irradiated with an accumulative dose of 210Gy, following a protocol of three pulses of 70Gy every 4 hr. Afterwards, worms were transferred to a new Petri dish with fresh f/2 and *ad libitum* food to avoid starvation. At 12, 24, and 72 hr post-irradiation (i.e. after the last irradiation exposure), RNA of 40 randomly collected worms was isolated. Prior to RNA isolation, worms were starved for 12 hr to avoid RNA contamination of diatoms. Control animals (n = 40) were cultured in the same way as the treated worms; however, the γ-ray exposure was omitted. Three independent replicates of irradiation were performed.

### Preparation and sequencing of RNA-seq libraries for de novo transcriptome assembly

RNA isolation

Worms were starved for 18–24 hr prior to RNA isolation to prevent diatom RNA contamination, then rinsed in fresh medium. Total RNA was extracted using TRIzol Reagent (Ambion, Foster City, CA), according to manufacturer's instructions. Animals were homogenized in TRIzol Reagent by pipetting. For every extraction, a batch of 200–300 worms was used. Samples were resuspended in nuclease-free water and treated with five daU of DNAse I (Thermo Scientific, Waltham, MA) for 45 min at 37°C. Enzyme and all the remaining DNA were removed by extraction with phenol:chloroform:iso-amyl alcohol (125:24:1, pH 4.5 Life Technologies, Waltham, MA). Samples were alcohol precipitated overnight at –80°C. Total RNA was pelleted by centrifugation at 12,000 g for 20 mins at 4°C, washed with 70% ethanol and air-dried for 5 min. RNA was resuspended in nuclease-free water. Concentration of total RNA samples was measured with Qubit RNA BR assay kit (Invitrogen, Waltham, MA).

Preparation of 454 library

Random-primed normalized cDNA library for 454 sequencing was prepared by Vertis Bioteknologie AG (Freising, Germany). Total RNA was isolated from the worms pellet using the mirVana miRNA isolation kit (Ambion). The RNA preparation was analyzed for its integrity by capillary electrophoresis. From the total RNA poly(A)+ RNA was prepared. First-strand cDNA synthesis was primed with a N6 randomized primer. Then, 454 adapters A and B were ligated to the 5' and 3' ends of the cDNA. The cDNA was finally amplified with PCR (16 cycles) using a proof-reading enzyme. Normalization was carried out by one cycle of denaturation and reassociation of the cDNA, resulting in N1-cDNA. Reassociated ds-cDNA was separated from the remaining ss-cDNA (normalized cDNA) by passing the mixture over a hydroxylapatite column. After hydroxylapatite chromatography, the ss-cDNA was amplified with 10 PCR cycles. For Titanium sequencing, the cDNA in the size range of 500–700 bp was eluted from a preparative agarose gel. An aliquot of the size fractionated cDNA was analyzed by capillary electrophoresis. The library was sequenced on GS FLX Titanium machine following manufacturer's protocol.

Illumina libraries

Poly(A)-tailed mRNA fraction was purified from total RNA and barcoded RNA-seq libraries were created using SureSelect Strand Specific RNA Library Prep Kit (Agilent, Santa Clara, CA) or NEXTflex Rapid Directional qRNA-Seq Kit (Bioo Scientific, Austin, TX) in accordance with manufacturer's protocol. Ribo-Minus depleted RNA was prepared by RiboMinus Eukaryote Kit for RNA-Seq (Ambion) and NEXTflex Rapid Directional qRNA-Seq Kit (Bioo Scientific). Sequencing was performed on the Illumina HiSeq 2500 machine.

### RAMPAGE libraries
5'-enriched libraries were prepared according to RAMPAGE protocol (*Batut et al., 2013*).

### 3'-specific libraries
The CEL-seq protocol (*Hashimshony et al., 2012*; *Junker et al., 2014*) combined with T-fill sequencing protocol (*Wilkening et al., 2013*) allows mapping of polyadenylation sites in mRNAs and thus precise annotation of 3' ends. While CEL-seq protocol is designed for single cells, it works well with larger amounts of RNA as well. For generating 3'-specific libraries RNA was extracted from whole adult sectioned worms using TRIzol reagent (Ambion), following the manufacturer's manual. RNA pellets were resuspended with barcoded primers consisting of a polyT stretch, a 4 bp random barcode, a unique sample-specific barcode, the 5' illumina adaptor, and a T7 promotor. The RNA samples were reverse transcribed, pooled, and in vitro transcribed for linear amplification with the MessageAmp II kit (Ambion) according to the CEL-Seq protocol (*Hashimshony et al., 2012*). Illumina sequencing libraries were made with the TRuSeq small RNA sample prep Kit (Illumina, San Diego, CA) and paired-end sequencing was performed using T-fill protocol as described in *Wilkening et al. (2013)*.

## Preparation and sequencing of RNA-Seq libraries from irradiated animals
At the defined time points after irradiation, animals were rinsed with fresh f/2 medium, suspended in 500 ul of TRIzol reagent (Ambion), and homogenized. Total RNA extraction was performed with the Direct-zol RNA MiniPrep kit (Zymo Research, Irvine, CA), according to the manufacturer's instructions. Concentration and quality of extracted RNA was measured using the Qubit RNA BR Assay Kit (Life Technologies). Total RNA was used to make RNA-Seq libraries with the SureSelect Strand-Specific RNA Library Prep kit for Illumina Multiplexed Sequencing (Agilent), in accordance with the manufacturer's protocol. Sixteen libraries were pooled (2 nM) and sequenced on the Illumina HiSeq 2500 machine.

## Preparation and sequencing of RNA-seq libraries from FACS-isolated cells
Cells were sorted from worms representing three conditions: intact worms, 1-day-old juveniles, and amputated heads. Intact worms were starved for 48 hr before sorting. To obtain the juveniles, newly hatched worms were collected and maintained for a day without food. To obtain heads, worms were relaxed in 7.14% $MgCl_2$, cut below the pharynx and let to regenerate for 24 hr without food. On the day of sampling, worms were collected in RNase-free 1.5-ml tubes. Worms were put on ice to facilitate aspiration of excess f/2, after which they were resuspended in 100 µl Otto1 buffer (0.1M Citric acid, 0.5% Tween in MilliQ) and incubated for 8 min at room temperature. 300 µl Otto2 (0.4M $Na_2HPO_4$ in MilliQ) buffer was added and worms were vigorously pipetted up and down to lyse them into a cell suspension. The suspension was diluted up to 1 ml by adding 600 µl 1:2 Otto1: Otto2, and samples were labeled with 4 drops of a live-cell Hoechst staining for 20 min at room temperature (NucBlue Live ReadyProbes Reagent, Thermo Fisher Scientific), after which samples were kept on ice.

For cell sorting, a gating strategy was developed including the elimination of cell debris and cell clusters, the identification of neoblasts, and finally the selection of G1 cells, representing mainly differentiated cells, and late-S/G2/M phase cells, representing proliferating cells (*Figure 1—figure supplement 4*). To sort the differentiated cells, a selection on the left side of the G1 peak was made to avoid contamination with early S-phase cells as much as possible. Sorting was performed using a Beckman Coulter Moflo Astrios (Central Flow Cytometry Unit, UMCG). Samples of 5000 cells per RNase-free tube were collected, which were put on ice. TRIzol reagent (Life Technologies) was added as fast as possible and the samples were stored at −80°C. RNA-Seq libraries were made using the CEL-Seq method and single-end sequenced using T-fill protocol as described above. For each studied condition three replicate libraries were generated.

For confirmation of specificity of the developed sorting strategy, six samples of 100 adult worms were collected, and three of them were irradiated with an accumulative dose of 210Gy as described above, while the other three samples were kept as controls. Worms were macerated and labeled

with Hoechst as described above. Flow cytometry was performed using a BD FACSCanto II, and the percentages of single cell Hoechst labeled cells within the 2C and 4C gates were determined with the FCS express software package.

## De novo transcriptome assembly and annotation

Raw RNA-seq reads were processed by the read cleaning module of Mira assembler v.4.0.2 (*Chevreux et al., 2004*) to trim adapter sequences and low-quality regions. Filtered Illumina paired-end reads were normalized by insilico_read_normalization.pl utility from Trinity v.2.0.6 package (*Grabherr et al., 2011*) to the maximum coverage of 30x and assembled by IDBA-tran v.1.1.1 (*Peng et al., 2013*), Trinity v.2.0.6 (*Grabherr et al., 2011*), and SOAPdenovo-trans v.1.0.4 (*Xie et al., 2014*) assemblers using strand information and default assembler parameters. For SOAP-denovo-trans assemblies with multiple k parameter were generated (k = 23,27,31,41,51,61). Non-stranded 454 data were assembled with Mira assembler v.4.0.2 (*Chevreux et al., 2004*). For each assembly, redundancy was removed using cd-hit-est program from CD-HIT v.4.6.1 (*Fu et al., 2012*) with parameters '-c 0.99 -T 0 -M 0', and the results of different assemblies were further merged using cd-hit-est with the same parameters. Next, RNA-seq reads were mapped back to the resulting contigs using Bowtie v.2.2.4 (*Langmead and Salzberg, 2012*), reads mapping to a given contig extracted and reassembled using CAP3 v 12/21/07 (*Huang and Madan, 1999*) and Newbler v.2.7 (*Margulies et al., 2005*) requiring consistent placing of paired-end reads. The reassembly pipeline is available at https://github.com/eberezikov/ReCAP. The reassembled contigs were merged with cd-hit-est and prioritized by the number of reads mapped. The primary set of contigs that explains 90% of all available RNA-seq reads was selected as the main assembly.

Transcriptome assembly quality was evaluated using TransRate v.1.0.1 (*Smith-Unna et al., 2016*) using polyA-enriched and Ribo-Minus depleted libraries. Homologs from human (GRCh37) and *S. mediterranea* (*Liu et al., 2013*; *Brandl et al., 2016*) were identified using blastx v.2.2.6 (*Altschul et al., 1997*) taking the best hits with e-value cutoff below 0.01. Pfam domains from Pfam database v. 27 (*Finn et al., 2016*) and Core Eukaryotic genes (*Parra et al., 2009*) were annotated using HMMER v.3.1 (*Eddy, 2011*). tRNA and rRNA genes were annotated using tRNAscan-SE v.1.23 (*Lowe and Eddy, 1997*) and RNAmmer v 1.2 (*Lagesen et al., 2007*), respectively. Assessment of transcriptome completeness was performed using BUSCO v.2 (*Simão et al., 2015*) with Eukaryota and Metazoa datasets.

Transsplicing leader sequence was identified by analyzing k-mer frequencies (k = 19) in the first 100 nt of transcripts. The consensus sequence CCGTAAAGACGGTCTCTTACTGCGAAGACTCAA TTTATTGCATG reconstructed from the overlapping frequent k-mers corresponds is the same as published previously (*Wasik et al., 2015*).

## Differential expression analysis of RNA-Seq data

RNA-seq reads were mapped to MLRNA1509 transcriptome assembly using Bowtie v.2.2.4 (*Langmead and Salzberg, 2012*), and gene-level counts for transcript clusters were calculated from the resulting bam files by Corset v.1.03 (*Davidson and Oshlack, 2014*) combining both irradiation and FACS datasets. Subsequent differential gene expression analysis was performed with edgeR package (*McCarthy et al., 2012*) separately for the irradiation and FACS dataset gene counts. Lowly expressed clusters were removed, requiring at least one count per million in at least three samples. Unwanted variation was removed with RUVSeq package (*Risso et al., 2014*) using k = 3 for the irradiation dataset and k = 1 for the FACS dataset. FDR cutoff of 0.05 was used for statistical significance.

## Whole mount in situ hybridization

cDNA synthesis was performed using the SuperScript III First-Strand Synthesis System (Life Technologies) according to the manufacturer's protocol with 2–3 µg of total RNA as a template for each reaction. Provided oligo(dT) and hexamer random primers were used.

DNA fragments selected as templates for in situ hybridization probes, were amplified from cDNA by standard PCR with GoTaq Flexi DNA Polymerase (Promega), followed by cloning using the pGEM-T vector system (Promega) and sequenced by GATC Biotech. All primers used are listed in *Figure 3—source data 1* and *Figure 3—source data 2*. DNA templates for producing DIG – labeled

riboprobes were amplified from sequenced plasmids using High Fidelity Pfu polymerase (Thermo Scientific). Forward (5'-CGGCCGCCATGGCCGCGGGA-3') and reversed (5'TGCAGGCGGCCGCAC TAGTG-3') primers binding the pGEM-T vector backbone near the insertion site were designed. Moreover, versions of the same primers with a T7 promoter sequence (5'-GGATCCTAATACGAC TCACTATAGG-3') appended upstream were obtained. The T7 promoter sequence serves as a start site in subsequent in vitro transcriptions. A pair of primers, depending on the orientation of the insert in the vector: forward with T7 promoter and reverse without or vice versa, was used to amplify every ISH probe template.

Digoxigenin (DIG) labeled RNA probes (500 to 800 bp in length) were generated using the DIG RNA labeling Mix (Roche, Switzerland) and T7 RNA polymerase (Promega, Fitchburg, WI) according to the manufacturer's protocol for in vitro transcription. The concentration of every probe was measured with the Qubit RNA BR assay (Invitrogen), probes were diluted in Hybridization Mix (*Pfister et al., 2007*) to 20 ng/µl, stored at −80°C and used within 4 months. The final concentration of the probe and optimal temperature used for hybridization varied for different probes and were optimized for each probe.

Whole mount in situ hybridization (ISH) was performed following an earlier described protocol (*Pfister et al., 2007*). Pictures were made using a standard light microscope with DIC optics and an AxioCam HRC (Zeiss, Germany) digital camera and the EVOS XL Core Imaging System (ThermoFisher).

## Fluorescent in situ hybridization and immunofluorescence

Fluorescent in situ hybridization (FISH) was performed following the published FastBlue protocol developed for planarians (*Currie et al., 2016*), except the 5% NAC treatment and bleaching steps were ommited. The primary, polyclonal antibody for Macpiwi1 (1:250) (*Wasik et al., 2015*) or the primary anti-phospho histone H3 Antibody (1:100) (Millipore, Billerica, MA) was added to the FISH antibody solution as 1:250. After FISH development, samples were incubated with secondary goat anti-rabbit IgG Antibody conjugated with FITC (Millipore), diluted 1:150 in blocking solution, for 1 hr at room temperature. Samples were then washed five times with PBS-T. Slides were mounted using 80% glycerol solution, and the labeling was visualized with a Leica TCS SP8 confocal microscope at the UMCG Imaging and Microscopy Center.

## RNA interference

In order to generate dsRNA fragments, the same plasmids were used as for making ISH probes. Templates for the synthesis of both sense and antisense RNA strands were amplified from the plasmids containing the insert of interest. The same primers were used as for ISH riboprobe template amplification, and for each fragment, two PCRs were performed – with both pairs of primers (forward with T7 promoter and reversed without and vice versa). High Fidelity Pfu polymerase (Thermo Scientific) in 150 µl of total volume reaction was used. PCR products were then run on 1% agarose gel, PCR product bands were cut out and purified using the QIAquick Gel Extraction Kit (Qiagen, Netherlands). Each template was then used to synthesize the corresponding single strand RNA with the TranscriptAid T7 High Yield Transcription Kit (Thermo Scientific) according to manufacturer's protocol. The single reaction volume was 50 µl, and tubes were incubated in 37°C for 5 hours. Afterwards 100 µl of nuclease-free water was added to each tube, sense and antisense RNA strands were mixed to a final volume of 300 µl and annealed by incubating them at 70°C for 10 min, followed by gradual cooling down to room temperature, taking approximately 90 min. Every sample was then treated with 1U of RNase A (Life Technologies) and 5U of DNase I (Thermo Scientific) for 45 min at 37°C. Samples were alcohol precipitated overnight at −80°C. dsRNA was pelleted by centrifugation at 12,000g for 15 min at 4°C, washed with 75% ethanol, and air-dried for 5 min. dsRNA was resuspended in nuclease-free water and the concentration was measured using Nanodrop ND1000. Freshly autoclaved and filtered f/2 medium was used to adjust the concentration to 10 ng/µl. Samples were aliquoted in 1.5 ml Eppendorf tubes and stored at −80°C.

Specific knockdown of candidate genes by RNA interference with double-stranded RNA delivered by soaking was performed as previously described (*De Mulder et al., 2009*). RNAi soaking experiments were performed in 24-well plates in which algae were grown. Individual wells contained 300 µl of dsRNA solution (10 ng/µl in f/2 medium) in which 15 individuals were maintained. RNAi

was performed for three weeks during which dsRNA solution was refreshed daily. Worms were weekly transferred to fresh 24-well plates to ensure sufficient amount of food. For each gene of interest, the effect on homeostasis and regeneration was studied. As a negative control, GFP dsRNA was used. In experiments addressing regeneration, the tail of worms was amputated after 1 week of RNAi. Photos of randomly selected worms were made 1 week after cutting for studying the effect of RNAi on regeneration, and between 2 and 3 weeks of treatment to study to effect on homeostasis.

### Mitotic labeling

Mitotic labeling was performed as described in *Ladurner et al. (2000)*. In short, both cut and intact worms were randomly selected at the 10th day of RNAi treatment (48 hr after amputation of the tail), washed in f/2 medium and relaxed in 1:1 MgCl$_2$:f/2 for 5 min, fixed in 4% paraformaldehyde (PFA) for 1 hr, washed with PBS-T (PBS and 0,1% Triton X-100) and blocked with BSA-T (1% bovine serum albumin in PBS-T) for 30 min. The primary anti-phospho histone H3 Antibody (Millipore) was diluted 1:100 in BSA-T and applied overnight at 4°C, followed by washing with PBS-T. Worms were incubated in secondary goat anti-rabbit IgG Antibody conjugated with FITC (Millipore), diluted 1:150 in BSA-T, for 1 hr at room temperature. After being washed with PBS-T, slides were mounted using Vectashield (Vector Laboratories US, Burlingame, CA). Mitotic cells were visualized using a Leica TCS SP2 confocal microscope and counted with the Cell counter plugin in ImageJ.

### Data accessibility

Web interface that provides search and visualization capabilities for the generated datasets is available at http://neoblast.macgenome.org. RNA-seq data have been deposited at DDBJ/EMBL/GenBank under the accession SRP082513. The transcriptome assembly has been deposited at DDBJ/EMBL/GenBank under the accession GEXL00000000. The version described in this paper is the first version, GEXL01000000.

## Acknowledgements

We thank ERIBA Sequencing Facility for performing Illumina sequencing and Marianna Bevova for assistance with implementing the T-fill protocol, Victor Guryev for fruitful discussions on data analysis, Kay van Nies for technical assistance with the project, Lisa Glazenburg for caretaking of *Macrostomum* cultures, Bret J Pearson for advice on the in situ hybridization protocol, and Gregory J Hannon and Kaja A Wasik for the kind gift of Macpiwi1 antibody. We are grateful to Alejandro Sánchez Alvarado, Jochen C Rink and an anonymous reviewer for the insightful comments and suggestions that helped to improve the manuscript. The inception of the study benefited from early discussions with Peter Ladurner and Lukas Schärer. This work was supported by the Horizon Breakthrough Grant from the Netherlands Genomics Initiative (grant no. 93511007) and European Research Council Starting Grant (MacModel, grant no. 310765) to EB.

## Additional information

### Funding

| Funder | Grant reference number | Author |
|---|---|---|
| Nederlandse Organisatie voor Wetenschappelijk Onderzoek | 93511007 | Eugene Berezikov |
| European Commission | 310765 | Eugene Berezikov |

The funders had no role in study design, data collection and interpretation, or the decision to submit the work for publication.

### Author contributions

MGru, SM, EB, Conception and design, Acquisition of data, Analysis and interpretation of data, Drafting or revising the article; DS, KdM, Conception and design, Acquisition of data, Analysis and interpretation of data; FB, MGre, WA, SvdE, Acquisition of data, Analysis and interpretation of data; PMW, Acquisition of data, Analysis and interpretation of data, Drafting or revising the article

## Author ORCIDs

Eugene Berezikov, http://orcid.org/0000-0002-1145-2884

# Additional files

## Major datasets

The following datasets were generated:

| Author(s) | Year | Dataset title | Dataset URL | Database, license, and accessibility information |
|---|---|---|---|---|
| Grudniewska M, Mouton S, Berezikov E | 2016 | Transcriptional signatures of somatic neoblasts and germline cells in Macrostomum lignano | http://www.ncbi.nlm.nih.gov/sra/SRP082513 | Publicly available at NCBI Sequence Read Archive (accession no: SRP082513) |
| Grudniewska M, Mouton S, Berezikov E | 2016 | Transcriptional signatures of somatic neoblasts and germline cells in Macrostomum lignano | https://www.ncbi.nlm.nih.gov/Traces/wgs/GEXL01 | Publicly available at NCBI Sequence Set Browser (accession no: GEXL00000000) |

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
