## [Decision Letter]

Thank you for submitting your article "Transcriptional signatures of somatic neoblasts and germline cells in *Macrostomum lignano*" for consideration by *eLife*. Your article has been favorably evaluated by Marianne Bronner as the Senior Editor and three reviewers, one of whom, Alejandro Sánchez Alvarado (Reviewer #1), is a member of our Board of Reviewing Editors. The following individual involved in review of your submission has agreed to reveal their identity: Jochen C Rink (Reviewer #3).

The reviewers have discussed the reviews with one another and the Reviewing Editor has drafted this decision to help you prepare a revised submission.

Summary:

The work reported by Dr. Berezikov and his team is important as it reports on a species different from *S. mediterranea* and occupying a different position in the platyhelminthes that should allow for comparative functional studies within this group of animals. The authors describe a transcriptomic approach and resources they have generated for studying stem cells (neoblasts) in the platyhelminth *Macrostomum lignano*. Specifically, the authors report on an impressive amount of sequencing data, and the use of irradiation and RNAseq approaches to define transcript sets expressed in gonads/germ line and or somatic stem cells. RNAi knock-down data and some in situs are presented to support the specificity of the gene lists identified.

Essential revisions:

There are a number of issues that the authors must address for this "Tools and Resources" contribution to be further considered for publication in *eLife*.

1) The authors need to make a better effort to both make the data accessible and amenable for inter-species comparisons. Otherwise the study loses its utility as a tool/resource. In this sense, the authors can get inspired from doi:10.1093/nar/gkv1148 and doi:10.1016/j.devcel.2015.11.004 and should implement similar tools as resource.

2) The authors need to provide metrics of the quality of the transcriptome in a more transparent fashion than currently done. The inclusion of transcriptome quality control metrics is essential, e.g. the relative frequency of chimeric, fragmented or redundant transcripts, or the fraction of transcripts from co-sequenced bacteria or algae. Such metrics are crucial to prospective users from outside the community to judge the quality of the limitations of the resource they are dealing with.

3) Additionally, the work would be of much greater general use and interest if the authors were to improve their comparisons and discussions between the stem cell/germ cell data reported for *M. lignano* to the many available datasets available for *S. mediterranea* stem cells and/or vertebrate stem cell resources. As it is currently, the presentation of the stem cell/germ cell data remains rather limited to *M. lignano*.

4) Controls are missing for the FACS profile analyses, in particular, a quantitative analysis between control and irradiated animals. Without this control, specificity of the findings remains questionable. The irradiation-derived stem cell data set contains "105 out of the 157 planarian neoblast genes", yet the corresponding FACS-derived stem cell data set contains only 26 out of 157 neoblast genes. Although the author's point that the FACS approach is necessarily limited to genes expressed in S/G2 is certainly valid, a high background of somatic cells in the FACS gates could also cause the low degree of overlap between the two data sets. The proposed control would go a long way to address this concern.

5) Finally, evidence for specificity of stem cell markers needs to be fortified. This could be done by either double in situ hybridizations of a few reported candidates with either piwi or by BrdU labeling.

---

## [Author Response]

*Essential revisions:*

*There are a number of issues that the authors must address for this "Tools and Resources" contribution to be further considered for publication in eLife.*

*1) The authors need to make a better effort to both make the data accessible and amenable for inter-species comparisons. Otherwise the study loses its utility as a tool/resource. In this sense, the authors can get inspired from doi:10.1093/nar/gkv1148 and doi:10.1016/j.devcel.2015.11.004 and should implement similar tools as resource.*

To increase the accessibility of the generated datasets we developed an online interface to this resource, which is available at http://neoblast.macgenome.org. The interface provides a straightforward way to search through the different transcript categories and to visualize and analyze the expression data of any gene of interest, for example by transcript ID, gene name or keyword. In addition, links to download the transcriptome assembly and the gene expression data and the classification into categories are provided. To facilitate the comparison of *M. lignano* and planarians, *S. mediterranea* homologs and their various classifications are provided, as well as links to PlanMine, which contains comprehensive information of planarian genomics (Brandl et al., 2015).

*2) The authors need to provide metrics of the quality of the transcriptome in a more transparent fashion than currently done. The inclusion of transcriptome quality control metrics is essential, e.g. the relative frequency of chimeric, fragmented or redundant transcripts, or the fraction of transcripts from co-sequenced bacteria or algae. Such metrics are crucial to prospective users from outside the community to judge the quality of the limitations of the resource they are dealing with.*

To assess the quality of the transcriptome assembly, we used TransRate – a recently developed reference-free approach that can detect common transcriptome assembly artefacts, such as chimeras and incomplete assembly, and provide individual transcript and overall assembly scores (Smith-Unna et al., 2016). The TransRate assembly score for the MLRNA150904 assembly is 0.4367, which ranks it as the 7^th^ highest scoring de novo transcriptome assembly out of 155 publicly available transcriptomes analyzed in Smith-Unna et al., 2016, and puts it into the top 5% of transcriptome assemblies by the overall assembly quality score. On the individual level, 51,990 out of 60,180 transcripts, or 86%, are classified by TransRate as ‘good’. The remaining 8,190 transcripts might have assembly errors but we decided to keep them in the assembly, since some genuine low-expressed transcripts might fall into this category. TransRate contig scores are included in the transcriptome annotation to facilitate transcript filtering as needed.

We have also added Benchmarking Universal Single-Copy Orthologs (BUSCO) assessment of the transcriptome, which shows a very similar distribution to the assessment of the published *Schmidtea mediterranea* transcriptome assembly Smed_dd_v6.

*3) Additionally, the work would be of much greater general use and interest if the authors were to improve their comparisons and discussions between the stem cell/germ cell data reported for M. lignano to the many available datasets available for S. mediterranea stem cells and/or vertebrate stem cell resources. As it is currently, the presentation of the stem cell/germ cell data remains rather limited to M. lignano.*

We have now added several planarian and mammalian datasets in the comparisons. Specifically, X1 neoblast dataset from Önal et al., 2012, gonad dataset from Wang et al., 2010 and Chong et al., 2011, sexual bias/specificity dataset from Resch et al., 2012, and mammalian pluripotency dataset from Tang et al., 2010. Overlaps between *Macrostomum* and these datasets are described in the new section in the Results (“Conservation of stem cell genes between *Macrostomum*, planarians and mammals”) and further discussed in the Discussion section.

*4) Controls are missing for the FACS profile analyses, in particular, a quantitative analysis between control and irradiated animals. Without this control, specificity of the findings remains questionable. The irradiation-derived stem cell data set contains "105 out of the 157 planarian neoblast genes", yet the corresponding FACS-derived stem cell data set contains only 26 out of 157 neoblast genes. Although the author's point that the FACS approach is necessarily limited to genes expressed in S/G2 is certainly valid, a high background of somatic cells in the FACS gates could also cause the low degree of overlap between the two data sets. The proposed control would go a long way to address this concern.*

We now included the requested experiment: “Irradiation of animals before sorting resulted in a six-fold decrease of the fraction of cells in the 4C gate (Figure 1—figure supplement 4), confirming that this gate represents proliferating cells and not contamination of e.g. doublets of differentiated cells.”

*5) Finally, evidence for specificity of stem cell markers needs to be fortified. This could be done by either double in situ hybridizations of a few reported candidates with either piwi or by BrdU labeling.*

To address this point, we have now performed double labeling of *Mlig-ddx39* with Macpiwi1 antibody and with phospho-histone H3 antibody: “A combined *Mlig-ddx39* FISH / Macpiwi1antibody labeling revealed cells co-expressing *Mlig-ddx39* and *Macpiwi1* in testes, ovaries and somatic neoblasts (Figure 5). Furthermore, a combined *Mlig-ddx39* FISH / phospho-histone H3 antibody mitotic labeling revealed expression of *Mlig-ddx39* in the proliferating cells of blastema (Figure 5). These observations provide additional evidence for neoblast-specific expression of *Mlig-ddx39.”*

We would like to note that unfortunately fluorescent in situ hybridization is not a robustly established technique in *M. lignano.* In fact, to our knowledge, Figure 5 is the first report of successful FISH in this animal. Unfortunately, we did not manage to optimize the protocol for genes other than *ddx39* in the time given for the revision. We reflect the current limitations of the method in the Discussion section: “[…] Therefore, it is important to develop more sensitive methods for visualizing gene expression in single cells in *M. lignano*, as this will be essential to confirm the specificity of genes for the somatic neoblasts or the germline. Our first attempts on implementing fluorescent *in situ* hybridization in *M. lignano* based on protocols developed for planarians (Currie et al., 2016) are encouraging (Figure 5) but it remains to be demonstrated how robust the method is when applied to a larger selection of genes.”